# Environmental Variables Related to *Aedes aegypti* Breeding Spots and the Occurrence of Arbovirus Diseases

**Adivânia Cardoso da Silva** [1] **and Paulo Sérgio Scalize** [2,*]

1 Graduate Program in Environmental and Sanitary Engineering, Federal University of Goiás, Goiânia 74000-000, Brazil; adivania.cardoso@hotmail.com
2 Post-Graduation Program in Environmental Sciences (CIAMB) and the Post-Graduation Program in Sanitary and Environmental Engineering (PPGEAS), Federal University of Goiás, Goiânia 74000-000, Brazil
* Correspondence: pscalize.ufg@gmail.com; Tel.: +55-629-8110-3030

**Abstract:** Outbreaks of dengue fever, Zika and chikungunya are realities that manifest themselves in almost the entire world. These are diseases whose main vector is *Aedes aegypti*. This is a synanthropic that needs three factors in order to survive: water, food, and shelter, which are available under favorable socio-environmental conditions. The goal of this work was to identify and evaluate the pertinence of environmental variables that can allow the growth of *Aedes aegypti* breeding spots and the eventual increase of dengue fever, Zika and chikungunya in rural areas. A systematic literature review was conducted by searching for works published in bibliometric databases, and the results were analyzed in qualitative and quantitative forms (statistical analysis). This search found 1007 works, of which 50 were considered pertinent to the subject. Each work was analyzed individually, and 16 distinct variables were found to be relevant and were afterward grouped into three categories: sanitation (SAN), climatology (CLIM) and socio-environmental factors, which are named "integrative" (INT) variables. The use of two or more categories was present in 52% of the works, highlighting the SAN + INT combination. Around 16% of the works have included rural areas, relating the arbovirus diseases in these regions to socio-economic and sanitation conditions. The paper analyses the impact of water characteristics on the growth of the mosquito, as well as on the increase of the three diseases. The monitoring of these 16 variables may allow for better arbovirus disease control and could be integrated into entomological vigilance programs for helping make decision processes concerning the prevention of diseases associated with water.

**Keywords:** predictor; environmental variables; environmental health; rural community; basic sanitation

## 1. Introduction

Arbovirus diseases such as dengue, Zika fever, chikungunya, mayaro and Yellow fever are of high incidence, especially in tropical or subtropical weather countries [1,2], although they also affect other countries [3,4].

The occurrence of arbovirus diseases has increased exponentially in recent years in several regions of the world, such as dengue in Saudi Arabia [5], the reappearance of its serotype 3 in Senegal [6], the high incidence of dengue, Zika and chikungunya in Brazil [7] and the dengue fever (FD) outbreaks in epidemic areas in China [8]. Due to the ongoing global phenomena, including climate changes and urbanization, the last-mentioned disease may increase the exposure risk of the entire population to over 60% by the year 2080 [9].

These arbovirus diseases are transmitted by *culicidae* of the *Aedes* (Stegomyia) genre, *Aedes aegypti* (Linnaeus, 1762), and *Aedes albopictus* (Skuse, 1984), which thrives among the human population due to three basic factors which guarantee its survival: water, food and shelter. Therefore, depending on the geographical location, the *Aedes* (Ae.) have a greater or shorter capacity of spreading the arbovirus, as evidenced by the predominance of the *Aedes albopictus* species ("Asian Tiger") in Asia, while in the Americas, the *Aedes aegypti* is the main vector [10].

However, the expansion of the *Aedes* has surpassed geographical barriers, reaching several regions of the globe (Figure 1), with an estimate of it being endemic in over 128 countries [11] as well as being present in other countries. As a consequence, people are infected by different types of arbovirus, increasing the arbovirus diseases prevalence in the world [12].

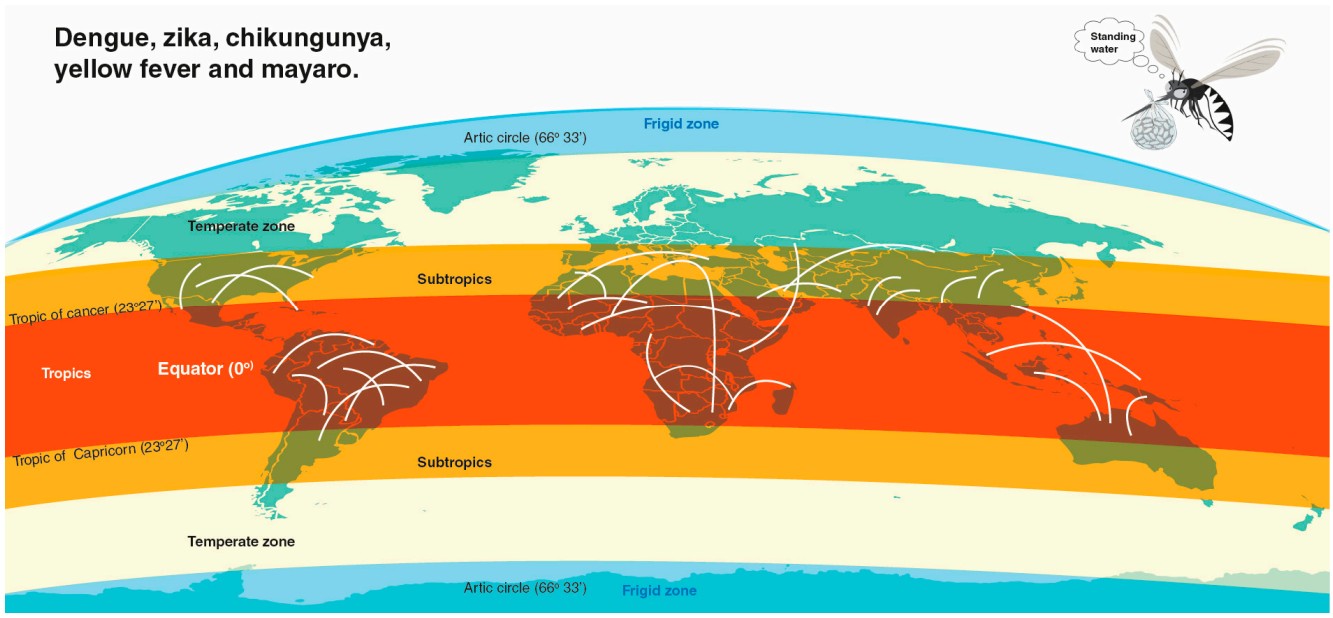

**Figure 1.** Presence of the *Aedes* in several countries throughout the world. Source: Drafted by the Authors. Note: The White lines illustrate the possible random routes of the *Aedes* vector from one place to another.

Therefore, the *Aedes aegypti* female (Diptera: Culicidae), after biting a person infected with one of the disease's serotypes, becomes a carrier that is able to transmit the virus to other people. It is a diurnal mosquito specimen, black-colored, with white stripes and spots. Its life cycle (Figure 2) comprehends four stages (egg, larva, pupa and adult form), which vary according to the number of existing larvae in the same breeding spot, the temperature and food availability [13]. Therefore, in order to reduce the vector's proliferation, the interruption of its life cycle via the elimination of breeding spots is necessary, and the aforementioned breeding spots [10,14,15] can be seen in places inhabited by humans [16,17].

Due to the intricate relationship between *Aedes aegypti* and the human being, the transmission of arbovirus in urban zones is favorable [18] due to the greater availability of spots for the vector's adaptation. However, in due course, the mosquito also adapts itself to the rural zone, be it due to the favorable environmental conditions [19,20] and/or the lack or precariousness of basic sanitation that allows the potential vector to create breeding spots and, consequently, spread diseases [21–23]. The *Aedes aegypti* breeding spots can take many forms, from solid waste discarded at random [24], water reservoirs for diversified use that do not go through a frequent cleaning routine (in general, supplying boxes, cans, water drums and drinking fountains) [11,15], as well as small flower vases in graveyards [25,26].

Water is fundamental for the vector's existence and the spreading of the disease, being one of the elements responsible for the maintenance of the vector Culicidae life cycle, while retained within potential different habitats for its proliferation, which generally are consequences of deficient basic sanitation. Thus, the places provided, for example, with water-supplying systems, would limit the irregular storage of water, reducing the vector population and the incidence of diseases transmitted by arthropods that thrive in the presence of water [27,28].

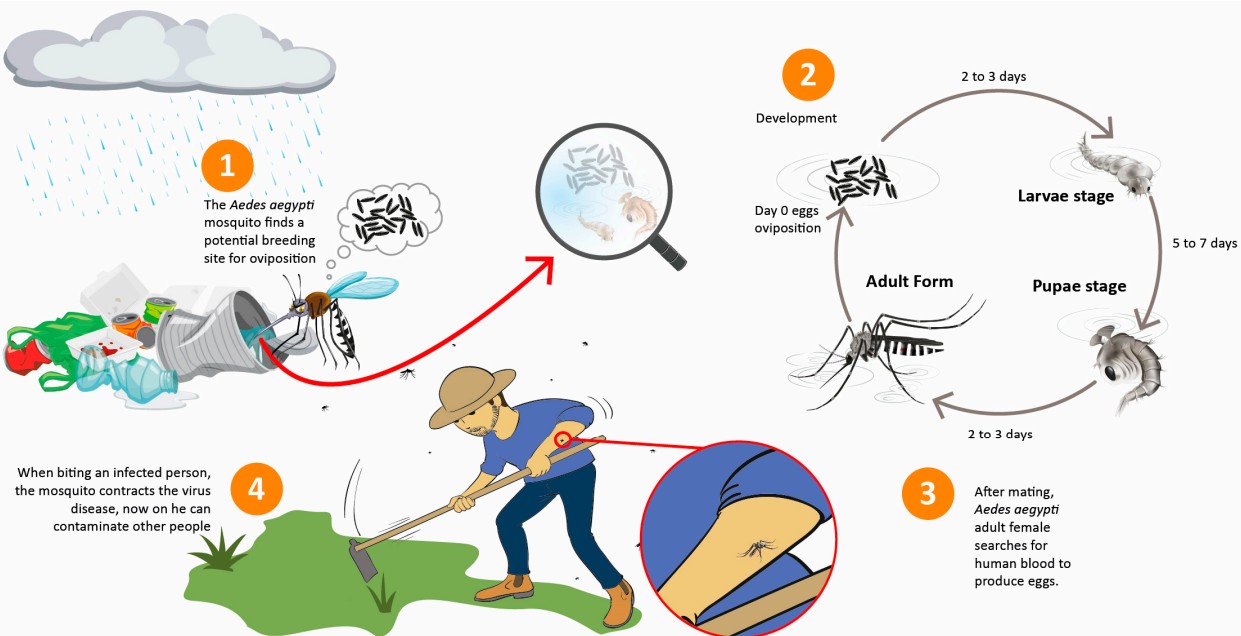

**Figure 2.** Life cycle of *Aedes aegypti* and potential breeding spots. Source: Drafted by the Authors.

Owing to the unavailability of an effective treatment for the emerging arbovirus diseases, such as a vaccine for dengue to be administered in national immunization programs, the affected countries face challenges for its prevention [29] which are in general, directed to the control of the *Aedes aegypti* and *Aedes albopictus* vectors.

There are several techniques that are used in fighting the *Aedes*, be it at the egg, larva and pupa (immature) stage or its adult stage. This can be done by the use of larvicidal products in mosquito breeding nests, aspersion of insecticide (interior/exterior), biological control, environmental management, traps, genetically modified mosquitoes, health education, community mobilization and participation, integrated interventions [30], as well as the use of natural vegetable products [31] and larvicidal made with kitchen oil as a substrate [32].

That way, knowledge about the variables which can influence the spreading of the mosquito throughout the world is of utmost importance, as it can be used in the proposition of preventive models [33].

In the literature, works pertaining to the environmental variables in association with the arbovirus diseases dengue fever, Zika and/or chikungunya are related mainly to diverse urban environment scenarios [34–40]. However, the discussion about the topic is growing, inferring that the same variables, or similar ones, favor the spreading of these diseases in rural environments as well due to the vector's presence, be it allied to environmental, basic sanitation and/or socio-demographic issues [41–43], among other factors. Consequently, there has been an increase in the last few years in the number of people who find these arbovirus diseases in rural environments [44,45], putting that population's health at risk. Therefore, it is necessary to know about the predictor variables that can imply the existence of breeding spots for *Aedes aegypti*, so that it becomes possible to apply more assertive measures for the sake of epidemiologic vigilance, prevention of arbovirus diseases and health promotion.

In this context, for a better understanding and exposition of those variables, it is possible to make use of the systemic literature review (SLR). The SLR refers to the technique of collecting, meeting, understanding, applying, analyzing, synthetizing and evaluating literature as support to a determined topic or method to be researched [46,47].

In this way, the goal of this work was to identify and evaluate the pertinence of the environmental variables considered in the literature as aggravating the existence of places

for the development of *Aedes aegypti* and/or occurrence of dengue, Zika fever and/or chikungunya in a rural area.

## 2. Materials and Methods

The research consists of five steps (Figure 3). The first step was dedicated to the surveying of technical-scientific works, with the choice of keywords which pertain to the research object using the *Strings*: (dengue OR zika OR chikungunya OR "*Aedes aegypti*") AND [(climatology OR "climate conditions" OR climate OR rainfall) OR ("basic sanitation" OR sanitation) OR ("solid wastes" OR waste OR trash OR garbage OR junk OR rubbish) OR (sewage OR "sanitary sewage" OR "domestic sewage") OR (water OR "water supply" OR "potable water") OR (drainage OR rainwater OR "rainwater management")], in the search system within the *Elsevier*'s *Scopus* database. The choice of keywords is based on the goal of reaching the largest possible amount of works which involved components of basic sanitation (water, waste, sewerage and drainage) and climatology (generally, meteorological elements) in association with arbovirus and *Aedes*.

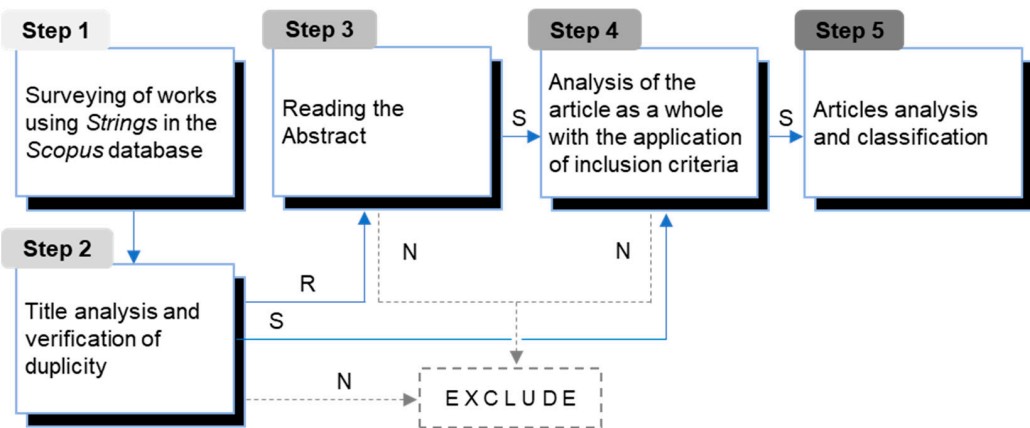

**Figure 3.** Flowchart of the steps of systemic literature review development steps in *Scopus*. Source: Drafted by the Authors. Note: S = Pertinent to the theme; R = review pertinence; N = Non-pertinent to the theme.

After surveying and tabulation, the works were analyzed without restricting them to aspects such as language, time series, book chapters, lecture works, editorials, letters), area (rural/urban) and country of study. Regarding the pertinence to the theme, they were classified as non-relevant (N), relevant (S) and review pertinence (R), where "S" were read in their entirety and "R" had their abstracts read to see if the works would be excluded along with "N" or not.

That way, in Step 2, from the surveyed productions selected in Step 1, the works were analyzed regarding their titles in order to exclude those inadequate to the theme (named "N", regarding socio-environmental variables not pertinent to arbovirus and vectors) and the duplicate/overlapping ones. The ones classified as "S" were immediately reserved for integral reading. Sequentially, in step 3, the works from step 2 classified as "R" were reviewed concerning their pertinence and analyzed through the reading of their abstracts to classify them as pertinent or not, as the mere title information did not clarify the research question. The ones tagged as "S" were sent forward to step 4, as those whose theme referred to the studies regarding association involving environmental variables, *Aedes aegypti* breeding spots and the prevalence or incidence of arbovirus diseases.

In the fourth step, each "S" work resulting from the third step went through a qualitative-quantitative analysis allied to the descriptive statistic, according to Snyder's [48] methodology, whose review is used to identify the evidence that is contained within the pre-established inclusion criteria to answer a determined research question. That way, the key verification elements were: (a) the vector is *Aedes aegypti* (yes; other, which one?); (b) what kind of arbovirus disease (dengue fever, Zika and/or chikungunya); (c) did it

consider the solid waste as *Aedes aegypti* breeding spots (no; yes, which ones?); (d) did it statistically treat the relationship among other basic sanitation components (water supplying, sanitary sewage or pluvial waters management); (e) did it consider one or more climatologic variable (such as temperature, rainfall, relative air humidity), if yes, (f) what was the analysis and components and/or variables? After tabulating these key analysis elements, the following inclusion criteria were applied: (i) being an association/correlation study or similar; and (ii) there must be a present association with environmental variables (sanitation, climate, socio-ecological and sociocultural). In this context, presenting a statistical analysis was fundamental to considering the research as pertinent.

Step 5 was a stage of re-analysis and detailing of the pertinent works resulting from Step 4. The works received classification upon confirmation of them possessing a set of data of the qualitative-quantitative information matrix that act as evidence for the key discussion elements. Thus, from the results found by the authors, the variables (and main indicators) were identified and separated into categories that demonstrated greater significance in the association with the vector *Aedes aegypti* allied to the incidence of dengue fever, Zika and/or chikungunya.

In addition, the statistical results found in the articles obtained were discussed in terms of the water variable regarding the incidence of arboviruses and breeding sites.

## 3. Results

The bibliographic research in the first step resulted in 1007 publications found. After analysis on the second step, 252 overlapped/doubled and/or thematically inadequate works were excluded, leaving 755 remaining works, of which 630 were excluded in the third step for not being studies regarding associations among environmental variables, *Aedes aegypti* breeding spots or other transmitter vectors such as *Aedes albopictus* and arbovirus diseases occurrence. In step 4, upon applying the inclusion criteria to the remaining 125 works, 50 were selected and 75 were excluded. Thus, in the fifth step, 50 articles were deemed suitable and qualified, as they had met the selection criteria regarding pertinence to the studied theme. Overall, 957 works were excluded from our review of the literature.

In the process of identifying the variables, it has been observed that in 1995, the productions began associating environmental factors with *Aedes aegypti* breeding spots and the occurrence of arbovirus diseases. Among the three diseases researched, the most cited one was dengue (and/or hemorrhagic dengue fever), which was present in 92.0% of the publications, while Zika and chikungunya were present in 20.0% and 22.0% of these, respectively. The *Aedes aegypti* is the most associated vector, as it was mentioned in 52.0% of the articles and cited along with *Aedes albopictus* in 26.0% of these. The remaining articles (22%) were not associated with a specific type of *Aedes*. Therefore, among the 50 analyzed articles, because of the specificities of each, 16 distinct variables were identified, which permitted their grouping into three categories as follows: sanitation (SAN), climatologic (CLIM) and integrative (INT) (Figure 4). The data collected can be found in Supplementary Materials Table S1.

In the SAN category, the variables "Water" (44%) and "Solid Waste" (38%) were the most used ones, and they represent basic sanitation, respectively, the components "Water supplying" and "Solid waste handling" were among the most used variables and, in similar percentage, it is noted that in CLIM, the variables "Rainfall" (42%) and "Temperature" (38%) are the meteorological elements of biggest significance considered by the authors in the analysis of the association with at least one INT category. In 97.1% (34/35) of the articles, the variables which compose the category were associated with one or more variables of the categories SAN and/or CLIM, with emphasis on the variables "sociodemographic" and "socioeconomic".

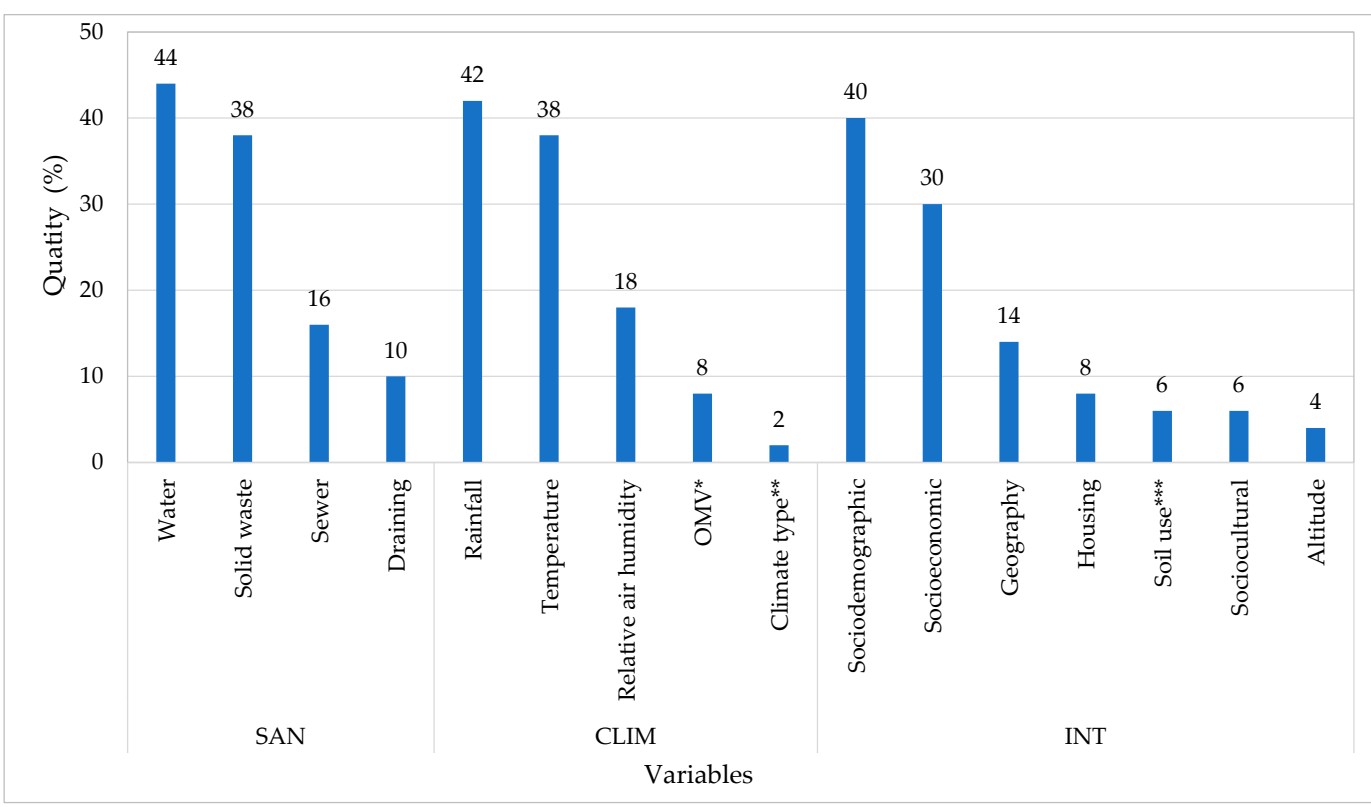

**Figure 4.** Division per categories of the variables found in the articles to which said variables were related to one or more arbovirus diseases and/or the research theme's vector. Source: Drafted by the Authors. Note: SAN = sanitation; CLIM = climatologic; INT = integrative; * other meteorological variables (wind speed, gust, sunlight hours, dew points and saturation deficit); ** tropical, subtropical and semi-arid.; *** urbanized area, agricultural area and grassland.

In Table 1, the main indicators per variable category can be observed, and such indicators were used in the publications regarding the present review's theme. It has been observed that 70 indicators were used by different studies which related them to health indicators.

**Table 1.** Division per categories of the variables with their main indicators considered in the literature as aggravating to the existence of *Aedes aegypti* and arbovirus diseases.

| Category | Variable | Indicator |
|---|---|---|
| Sanitation (SAN) | Water (AG) | AG1) Proportion with piped AG; AG2) Scarce access to city AG infrastructure; AG3) Proportion of permanent particular households whose AG supplying method does not pass through the city water system; AG4) Presence of recipient with AG and larvae (tires, pots, flower vases and barrels). |
| | Solid Waste (RS) | RS1) Percentage of RS collecting by cleaning service; RS2) Common and selective RS collecting; RS3) Proportion of permanent private households in which the RS is burned, buried, thrown into an empty lot or patio, river, lake, sea or other destination; RS4) Piling of solid waste in the household's surroundings; RS5) Existence of RS spread or piled in the peridomicile. |
| | Sewage (EG) | EG1) Rate of households with EG system; EG2) Proportion of households with skeptic tank; EG3) Households without EG service. |
| | Draining (DR) | DR1) Amount of households with drainage network; DR2) absence or not of Rainwater collection system; DR3) Presence of stagnant water within internal DR holes. |

**Table 1.** *Cont.*

| Category | Variable | Indicator |
|---|---|---|
| Integrative (INT) | Sociodemographic (SD) | SD1) Demographic Density; SD2) Percentage of people residing in the rural zone; SD3) Number of family members in the house; SD4) Education level; SD5) Sex; SD6) Color/race; SD7) Age; SD8) Marital Status; SD9) Occupational situation; SD10) Residence time; SD11) Influx of people. |
| | Socioeconomic (SE) | SE1) Per capita monthly income; SE2) GDP per capita; SE3) Unemployment rate; SE4) Gini Index; SE5) Percentage of people vulnerable to poverty; SE6) Human Development Index; SE7) Electricity. |
| | Housing (HB) | HB1) Building type; HB2) Screened doors or windows; HB3) Gutter; HB4) Number of rooms; HB5) Use of mosquito net, insect repellent and fumigation inside the house. |
| | Geography (GE) | GE1) Distance of the household top laces with high potential for larval density (graveyards, landfill sites, etc.); GE2) To the closest water bodies or courses. (excluding sea), GE3) to dairy farms, GE4) to other houses, GE5) to tire shops; GE6) to plant nurseries and GE7) abandoned sites. |
| | Soil usage (US) | US1) Urbanized area, forested and open agricultural; US2) Biome type; US3) Vegetation density; US4) Estimated vegetal coverage by satellite image (MODIS and NASA's Landsat). |
| | Sociocultural (SC) | SC1) History of family members travels; SC2) Frequency of recipient cleaning and water reservoir protection; SC3) Practices regarding water recipient storage, trash disposal and health education; SC4) Knowledge, perceptions, attitudes and community practices in the prevention of diseases (ACP). |
| | Altitude (ALT) | ALT1) Average altitude elevation (Above sea level). |
| Climatologic (CLIM) | Temperature (TE) | TE1) Average air temperature, annual maximum and minimum; TE2) Dew point temperature; TE3) Terrestrial surface TE. |
| | Precipitation (rainfall) (PR) | PR1) Annual rainfall precipitation average; PR2) Estimated rainfall (CHIRPS); PR3) Rainy days; PR4) Great rain events. |
| | Air humidity (UR) | UR1) Relative air humidity (%). |
| | Climate (CL) | CL1) Climate type (tropical, subtropical, semi-arid). |
| | Other Meteorological variables (OMV) | OMV1) Pressure surface; OMV2) Wind speed; OMV3) wind direction; OMV4) Wind gust; OMV5) Dew point; OMV6) Saturation deficit; OMV7) Sunlight hours. |
| **Total of indicators** | | **70** |

Source: Drafted by the Authors.

In Table 2, there is a presentation of the health indicators used in publications concerning the association with the different indicators from categories SAN, CLIM and/or INT. The number of notification cases (50%) was the most used data in the aforementioned presentation, followed by data regarding positivity (26%) for the arbovirus diseases dengue fever, Zika and chikungunya, with or without indication of a viral investigation method application in the article, and the use of incidence data happened in 10% of the works. The remaining 14% of the publications did not include correlations with the arbovirus diseases, but to the transmission vector's *Aedes aegypti* and/or *Aedes albopitcus* breeding spots.

In Figure 5, it is possible to see the number of surveys that were carried out in each country where the relationship with environmental variables was observed. In Figure 6, highlighting Brazil at the left side (the country with the most amount of work produced in this scope), the publications are presented in a timeline along with the distribution per the categories. It was also possible to observe the productions that have focused concomitantly on rural and urban areas 16% (8/50), exclusively urban areas 44% (22/50), involving peripheral areas, and the ones that did not mention the local of the study 40% (20/50).

**Table 2.** Health indicators used in the literature as variable answers in correlation with indicators of the SAN, INT and CLIM categories.

| Indicator | Recurrence (%) |
|---|---|
| Notified cases of dengue, Zika and/or chikungunya | 50 |
| Incidence of dengue, Zika and/or chikungunya | 10 |
| Positivity for dengue, Zika and/or chikungunya/detection through IgG/IgM/ELISA, NS1 and/or RT-PCR serology | 20 |
| Positivity for dengue, Zika and/or chikungunya/ * | 6 |
| Did not make correlations with arbovirus diseases but with breeding spots of the *Aedes aegypti* and/or *Aedes albopictus* transmission vector | 14 |
| Total | 100 |

Note: IgG and IgM: positivity for antibodies of the immunoglobulin G type (Class G) or immunoglobulin M (Class M); ELISA = Enzyme-linked; NS1 = Non Structural; RT-PCR = Reverse Transcriptase reaction followed by polymerase chain reaction; * does not mention viral marker. Source: Drafted by the authors.

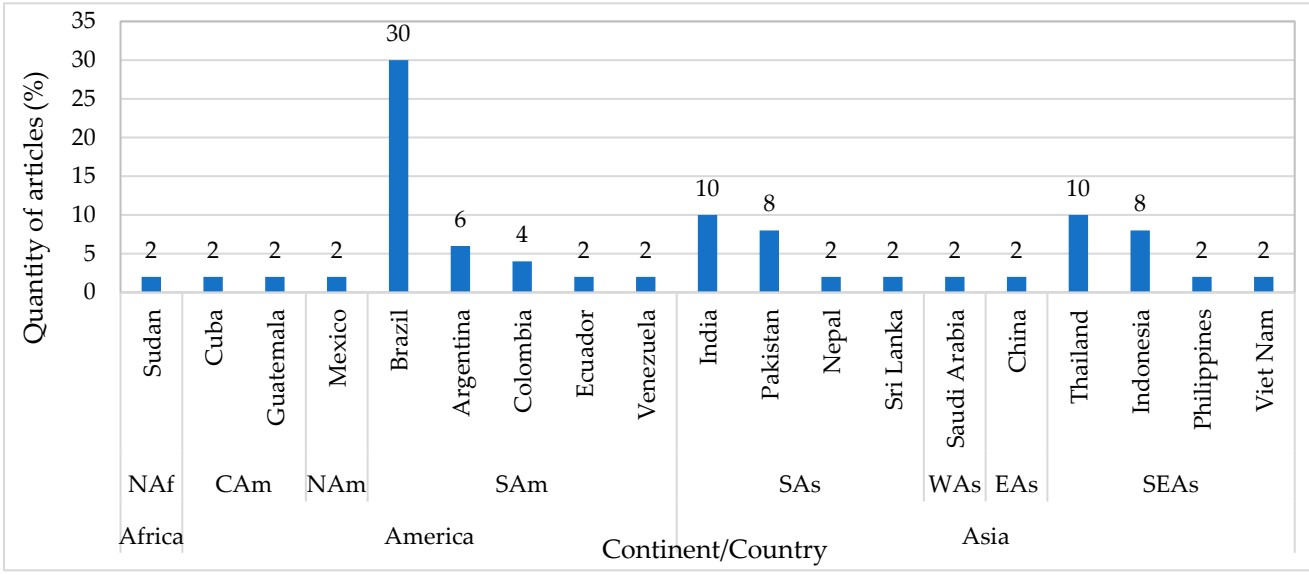

**Figure 5.** Distribution by country of searches found. Note: North Africa = NAf; Central America = CAm; North America = NAm; South America = SAm; South Asia = SAs; West Asia = WAs; Eastern Asia = EAs; Southeast Asia = SEAs.

Regarding the recurrence of categories, it is noted in Table 3 that INT was researched in 68% of the articles, followed by SAN in 66% and CLIM in 54%. The sum goes beyond 100% since 30% worked with a single category, 52% worked with two categories, and 18% worked with all three categories simultaneously.

The variables of a determined category were treated within the category itself (isolated) or in combination with the variables of one or more categories (jointly). Thus, when treated singly, CLIM was the most recurrent (24%) followed by SAN (4%) and INT (2%). In the tandem treatment of one or more categories, the combination SAN+INT was present in 40% of the works, followed by SAN+CLIM+INT (18%), CLIM+INT (8%) and SAN+CLIM (4%).

Analyzing the number of variables used in the same study (Figure 7), the greatest amount was that of nine, which presents 56% of said variables (9/16) as the aforementioned situation found in a single publication. The least number of variables found in the same article was one, which was observed in 8% (4/50) of the articles that had been analyzed.

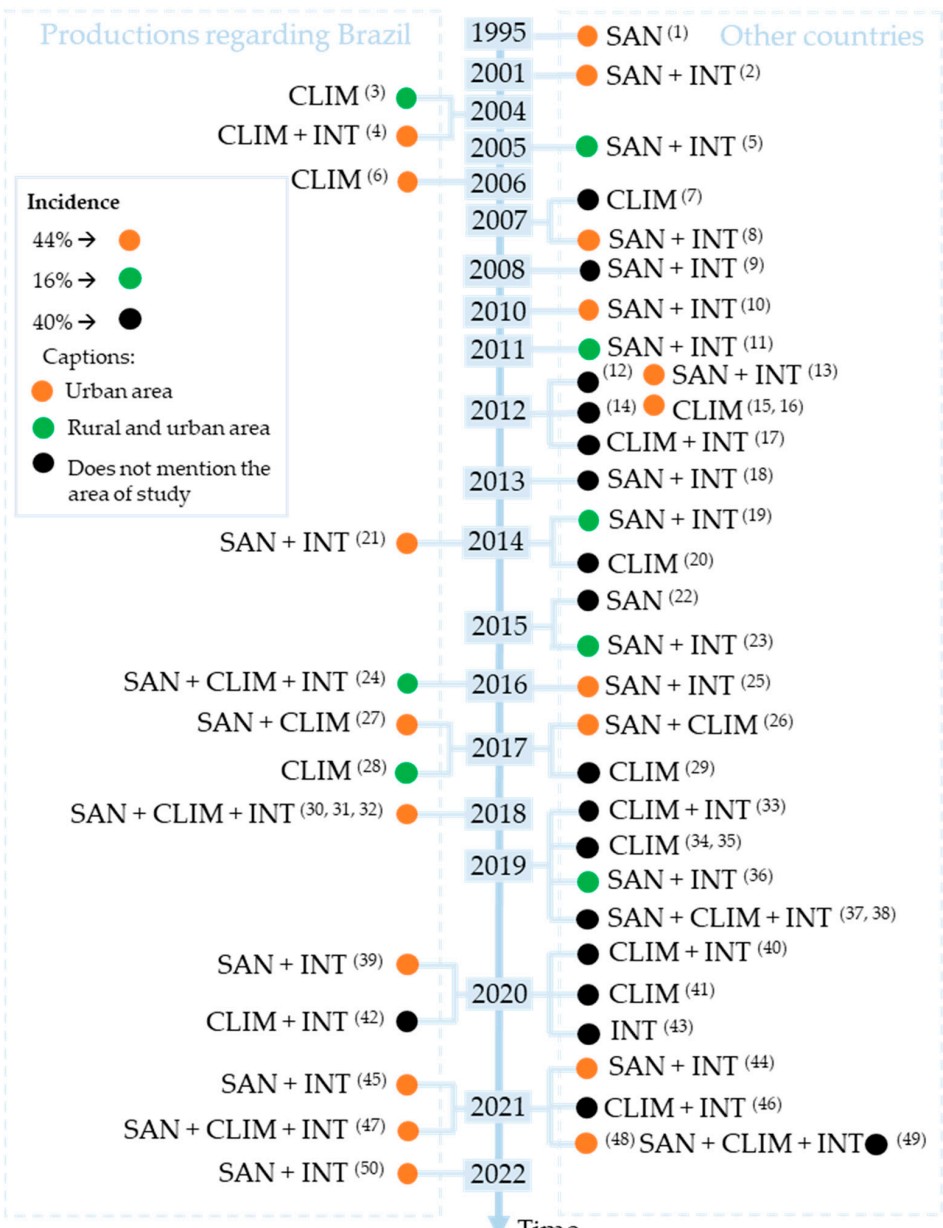

**Figure 6.** Timeline of publications found related to environmental variables and arbovirus diseases, in the *Scopus* database in 2022. Source: Drafted by the Authors. Note: (1) = Barrera et al. [49]; (2) = Bohra and Andrianasolo [50]; (3) = Gonçalves Neto and Rebêlo [51]; (4) = Penna [52]; (5) = Thammapalo et al. [53]; (6) = Favier et al. [54]; (7) = Nitatpattana et al. [55]; (8) = Brunkard et al. [56]; (9) = Nagao et al. [57]; (10) = Peraza et al. [58]; (11) = Anish et al. [59]; (12) = Kholedi et al. [60]; (13) = Mukhtar, Salim and Farooq [61]; (14) = Rana, Latif and Akhtar [62]; (15) = Wai et al. [63]; (16) = Chen et al. [64]; (17) = Carbajo, Cardo and Vezzani [65]; (18) = Thakolwiboon et al. [66]; (19) = Suwannapong et al. [67]; (20) = An and Rocklöv [68]; (21) = Honorato et al. [69]; (22) = Ballera et al. [70]; (23) = Siregar et al. [71]; (24) = Rodrigues et al. [72]; (25) = Delmelle et al. [73]; (26) = Kenneson et al. [74]; (27) = Fuller et al. [75]; (28) = Correia Filho [76]; (29) = Malik et al. [77]; (30) = Silva and Machado [78]; (31) = Aguiar et al. [79]; (32) = MacCormack-Gelles et al. [80]; (33) = Ishak, Sartika and Darmawanyah [81]; (34) = Ngweta et al. [82]; (35) = Tuladhar et al. [83]; (36) = Madewell et al. [84]; (37) = Siregar and Makmur [85]; (38) = Mala and Jat [86]; (39) = Mol et al. [87]; (40) = Sintorini, Aliyyah and Sinaga [88]; (41) = Shabbir, Pilz and Naeem [89]; (42) = Silva et al. [90]; (43) = Elaagip et al. [91]; (44) = Roy et al. [92]; (45) = Costa et al. [93]; (46) = Morgan, Strode and Salcedo-Sora [94]; (47) = Raymundo and Medronho [95]; (48) = Sánchez-Díaz et al. [96]; (49) = Telle et al. [97] and (50) = Ferreira et al. [98]. [reference] [49–98].

**Table 3.** Recurrences of the SAN, CLIM and INT categories, individual and combined incidence.

| Categories | Recurrence | Incidence |
|---|---|---|
| INT | 68 | 2 |
| SAN | 66 | 4 |
| CLIM | 54 | 24 |
| SAN + INT | - | 40 |
| SAN + CLIM + INT | - | 18 |
| CLIM + INT | - | 8 |
| SAN + CLIM | - | 4 |
| Total | 100 | 100 |

Note: Not applicable (-). Source: Drafted by the authors.

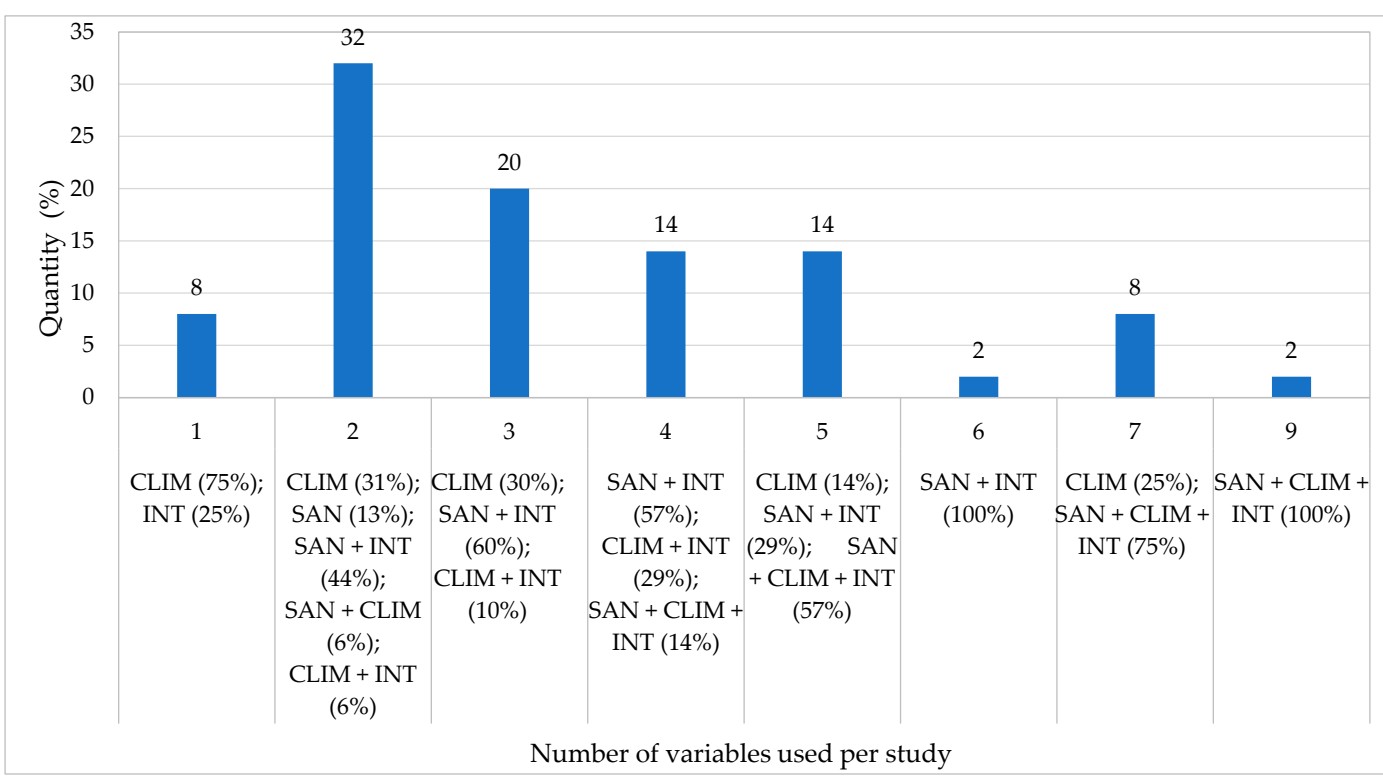

**Figure 7.** Division per incidence of the variables' categories according to the quantity used in a same article.

The use of more than one variable in the same study is incipient since 32% of the studies have used two variables, where 31% of these referred to CLIM and 13% to SAN, while the remaining (56%) used a variable of each category in three combinations, where SAN+INT (44%) was the most expressively used.

However, observing Figure 7 in a parallel analysis with the timeline (Figure 6), it is been noted that from 1995 to 2015, none of the 23 publications have made use of the three categories concomitantly, and 39% used a single category, CLIM (30%) and SAN (9%). On the other hand, 61% used two categories jointly, which are the combination CLIM+INT used in only one study and, expressively, SAN+INT in 57% of the analyzed articles, where 22% of the articles involved a rural zone in this first period. Since 2016, the number of articles using only one variable category has been reduced by 33%. Therefore, of the 27 works analyzed, only two categories had isolated use, CLIM (19%) and INT (4%), and 78% used at least two categories in the same study, where the combination SAN+CLIM+INT (33%) was the most used, followed by SAN+INT (26%), CLIM+INT (11%) and SAN+CLIM (7%), and 11% of the articles included the rural zone.

Therefore, in spite of the existence of an increase in the use of more than one variable of distinct categories in studies using this theme, there is a demand for research involving, besides the rural zone, a greater number of variables analyzed jointly, which could better explain the dissemination or not of arbovirus diseases in a given location. From 2016 to 2022 (Figure 6), where there was a growth of 33% in the number of publications that considered two or more variables categories in their research, the three categories (SAN, CLIM and INT) had a similar presence in 100% of the works, with emphasis on CLIM having more isolated presence in analyses. This must be due to the great importance of considering the climate in the investigation of diseases transmitted by vectors, according to the recommendation made by the World Health Organization (WHO).

## 4. Discussion

### 4.1. Significant Relationship between Water and Aedes Breeding Sites

Among those analyzed, 44% (22/50) dealt with the relationship between water and arboviral diseases and/or breeding sites for Culicidae vectors (Figure 4 and Supplementary Materials Table S1). The works were distributed equally in the countries of America (50%) and Asia (50%), specifically in the South and Central America regions, and South and Southeast Asia, where more 1.54 million people were involved in the studies since they had been infected with at least one of the arboviruses (dengue, Zika and/or chikungunya). Expressively, dengue was analyzed in 64% of the works, followed by chikungunya (18%) and Zika (18%), with 9% of them addressing at least two arboviruses, and 18% (4/22) dealing with mosquito breeding sites of the genus Aedes in association with water. However, they were statistically related to the incidence/prevalence of arboviruses.

In this way, the indicator analyzed: AG1 (proportion with piped water), AG2 (presence of recipient with AG and larvae—tires, pots, flower vases and barrels), AG3 (scarce access to city AG infrastructure) e AG4 (proportion of permanent particular households whose AG supplying method does not happen to pass through the city water system) were used in 45%, 14% (AG2/AG3) and 36% of the works, respectively. The sum exceeds 100%, as 9% analyzed two indicators simultaneously.

Thus, the theme "water" was discussed as an important indicator for the proliferation of *Aedes aegypti* and/or *Aedes albopictus* vectors which, directly and/or indirectly, contribute to the greater or lesser spread of arboviruses transmitted by them.

The abundance of vectors was identified in places with the presence of containers that contained water [49,53,59,66,84,92], either for watering animals (in general, drinking fountains) or for various uses (drums, water tanks, etc.). This scenario is due to the irregularity in the availability of water, which is a significant factor in the associations, both with the reproduction of the vector and for the occurrence of the disease. A study revealed that the interruption in the water system was related to the abundance of the vector ($p < 0.01$) [49] and the occurrence of dengue ($p < 0.05$) [74], as it was 3.80 times more likely to reach a person without a regular water supply [97], compared to a 3.46 times chance of not getting infected in a scenario of regular infrastructure [71].

In addition, precarious access to water for potability purposes encourages the population to adhere to alternative water sources (generally, individual/private wells), which was observed in 14% (3/22) of the studies [57,69,98] in regions of America and Asia, whose modality favored the increased risk of disease incidence ($p < 0.001$).

In this sense, the theme of water permeates among research questions of notable relevance to public health, especially in the context of basic sanitation.

### 4.2. Sanitation

An extensive discussion regarding basic sanitation has been observed in the articles, as this was present in 66% of the publications. In the period between 1995 and 2015, during which no publication used the three categories altogether, SAN was the most frequently used, in relation to CLIM and INT. This must be due to the tight relationship between sanitary conditions, the reproductive cycle of the *Aedes aegypti* and the human

being, as corroborated by the literature [27,99]. Thus, in Venezuela, there was an analysis of the correlation between a deficit in the water-supplying systems and the proposal for solid waste collection educational campaigns in environmental health aimed at reducing *Aedes aegypti* breeding spots [49]. In this scope, several authors list a variety of places that could facilitate the vector's proliferation, such as the ones verified by Thammapalo et al. (2005) [53], recipients such as containers and water pots for diverse uses often discarded, tires, water reservoirs and flower vases, that which was commonly observed in the peridomicile; unprotected tanks containing water [58,100]; random discarding of trash near houses, and diverse rubbish types that can accumulate water [70,84].

Disposable waste, usually recipients made of plastic and glass, when mishandled (left indiscriminately in open and shaded environments), are favored places for the larval development of *Aedes* mosquitoes [101,102], and these clusters of waste are commonly seen in rural communities of Goiás [103–105], in which the mosquito development situation is generally favored, because of the lack of coverage in basic sanitation services and even due to the practice of discarding agricultural waste or of revenue-generating activities, which can benefit the vector's spreading, although it does not necessarily benefit the disease spreading per se. However, in rural areas, mainly due to the lack of water supply and proper handling of domestic waste, the dissemination of dengue may happen at least as much as it occurs in cities [45,106], in spite of the fact that the transmission risk is focused on urban sites due to the greater population density [43,72].

It is possible to observe that the potential breeding spots for *Aedes aegypti* are presented in parallel association with sanitation, notably with regard to the components of water and solid waste, which corroborates several research works within this scope. In the Brazilian SE region, the rates of dengue incidence have been associated with the management of solid waste [87]. In Indonesia, the frequency of handling solid waste and the existence of adequate drinkable water facilities, as well as the education and community knowledge about dengue have had an important role in the reduction of its frequency [71]. Naturally, the population facing the inexistence or deficit of access to better disease prevention measures, such as vaccines, but having information that aims towards their own protection, makes it possible that measures are taken to reduce the number of vectors in their households, as such are good practices in the prevention of diseases that results in health promotion, and in addition, reducing the amount of mishandled water vessels and replacing these for trustworthy water-supplying systems as well as reducing the number of vectors [28,33].

### 4.3. Integrative

Given that one of the forms for the prevention of arbovirus diseases is to mitigate the pathogenic agent's transmission, the community has a crucial role in it, and that is a topic discussed by several authors upon associating it with the incidence of diseases and environmental variables. Kenneson et al. [71], in Ecuador, verified that actions that involved knowledge, attitudes and practices (KAP) were considered preventive to reduce the risk of arbovirus diseases, which is corroborated by several authors [59,60,66,67,107–109] who considered health education allied to good practices in the household environment to be a significant factor regarding prevention and control of arbovirus diseases.

That way, in the literature, several authors discuss that socio-environmental factors influence directly in health and show such correspondence involving several variables. According to investigation made by Silva and Machado [78] upon associating dengue and socioenvironmental variables in Brazil, the results suggested that the amassing of solid wastes, deficit in water supplying and sanitary sewage systems, allied to meteorological variables (such as temperature, rainfall and relative air humidity) presented relation to the existence of the vector. Rodrigues et al. [72] upon analyzing the temporal and spatial evolution of dengue in Brazil between 2001 and 2012, verified that the cases focused on the state of São Paulo and semi-arid regions, inferring a link with the basic sanitation services (waste retrieval and sanitary sewage), type of climate and sociodemographic factors.

Therefore, variables frequently cited in several works were the socioeconomic correlated to demographic factors, sanitary, and environmental conditions. Costa [93] have found a directly proportional relation between socioenvironmental, economic and demographic factors for the incidence of dengue, Zika and chikungunya in the Brazilian NE region. Regarding dengue, Honorato [69] researched a connection in sociodemographic variables and verified that inadequate trash retrieval and per capita income of up to 3 minimum salaries are both extrinsic factors to the health sector, which imply in greater or smaller risk for the disease, similar to what was found by Raymundo and Medronho [95], where the average income of 1 to 2 minimum salaries was considered a risk factor for the occurrence of Zika in the SE region of the country. MacCormak-Gelles [80], when analyzing the epidemiological and determining characteristics of dengue transmission in Fortaleza (Brazil), inferred a strong association with poverty and irregular retrieval of solid waste during low-transmission periods. That way, in the literature, several other authors [52,56,57,61,73,91,94,97,98] corroborate the question of where socioeconomic and/or demographic variables, sometimes allied to environmental factors, interfere for smaller or greater incidences of vector-transmitted diseases.

Thus, the incidence of arbovirus diseases can be aggravated in communities of low economic level [110] in areas with supply irregularities and households with precarious structures [111]. However, in communities with good solid waste disposal conditions, there is a strong potential for the reproduction of culicidae, which is allied to the care of water storage and access to information about prevention. As such, populations may be less stricken, since these practices mitigate disease transmissions due to vector control in the household environment.

In that sense, anthropogenic interferences in land, air, etc., have been emphasized in a work regarding the association between vectors and arbovirus diseases, allied to the theme of basic sanitation. Aguiar et al. [79], in Brazil, upon verifying the potential risk for Zika and Chikungunya outbreaks inferred that the variable of "land usage" (urban area, forested area and open agricultural) better defined the arbovirus diseases distribution, with the variable "solid waste" being one of the most significant for the existence of *Aedes aegypti* breeding spots. The changes in use and land coverage may contribute to vector dissemination, such as deforestation caused by anthropic action [112]. The inadequate destination for solid waste generated in an area with buildings, for example, may be a reason for the dissemination of vector-transmitted diseases in comparison to smaller incidences in regions with a greater forest area [113].

Despite how several studies consider the incidence of arbovirus diseases to be restricted basically to the urban environment, more distant areas or predominantly rural areas have been studied in the analysis of potential breeding spots of cullicidae. A study in Guatemala evidenced a greater abundance of *Aedes aegypti* larvae and pupae upon examining associations between the proximity of houses, roads and domestic environmental factors, which is named in the study as "environmental capital" (access to sanitation and better life conditions regarding household attributes: household standards, electricity and TV) [84]. Bohra and Andrianasolo [50], upon applying the Geographical Information System (GIS) in the modeling of dengue risk in semi-urban areas in India have revealed that sociocultural, cultural and socioeconomic data highlighting household standards, protection measures against the vector, cultural practices revolving around water storage and frequency in the supplying and waste collection have all contributed significantly to the incidence of dengue, with a $R^2$ of 0.958. Within this same theme and methodology, Roy et al. [92] have determined the inter-relationship for the incidence of dengue in the city of Jaipur and found out that sociocultural and economic factors influence directly the dissemination of the disease.

The *Aedes aegypti* is a mosquito of domestic presence. Thus, regardless of if the household is in a rural or urban area, if there are vector foci and environments which enhance its reproduction, such as unprotected water reservoirs and long periods between cleaning routines, etc., allied to the presence of an infected person, which may originate

from the urban perimeter, the rural population becomes more vulnerable to the transmission of the arbovirus.

*4.4. Climatologic*

It has also been noted that meteorological conditions had a frequent and significant presence in correlation analysis regarding arbovirus diseases, where several authors [51,55,62–64,76,77,82,89,114–116] have registered their results in the literature regarding such a theme. This can be due to the mosquito undergoing environmental conditioning in terms of preference for one region because of climate conditions for its better adaptation, as referred to by Favier et al. [54] when inferring that the vector's life cycle is related to both the human environment and the climate. This is corroborated by Silva et al. [90], where the authors found a positive correlation between climatology, dengue, chikungunya and solid wastes.

Therefore, regarding the *Aedes aegypti* oviposition dynamics, meteorological factors were also analyzed in the literature. As investigated by Sánchez-Díaz et al. [96], in Argentina, when identifying the night temperature and, in a context-dependent manner, the rainfall, vegetal coverage and demographic density influence the vector's proliferation in a given region. Tuladhar et al. [83], upon evaluating the effects of meteorological factors in the seasonal prevalence of vectors in the mountain range and plains regions in Nepal, revealed that the factor temperature-rain and relative humidity contributed to the *Aedes* vectorial indexes. In a study conducted by Siregar and Makmur [85], while investigating the transmission of dengue hemorrhagic fever in Indonesia, it was verified that the disease suffers influence of climate conditions, such as rainy days and average and maximum temperatures, as the *Aedes* vector is sensitive to the climate. In the context of climate change, due to the persistence of the planet's warming, which results mainly from human activity, the health sector has an enormous challenge to face, as alerted by the International Panel about Climate Changes (IPCC). Climate changes can influence the dissemination of vectors, causing migration to different regions, which would consequently increase the incidence of diseases by amplifying their geographical distribution [40,117].

It is noteworthy that, if at a given region the meteorological elements represent a factor which limits the existence of the *Aedes aegypti*, it is logical to consider the variables of temperature and rainfall in the investigation of potential breeding spots, since the climate condition may contribute for the vector index [83] and when associated to a complexity of social-environmental and/or socioeconomic factors, etc., such as the absence or scarcity of sanitation, poverty or even deficit in environmental education practices, the literature summons us to involve the community in order to join forces and confront the challenges that cover the prevention of diseases transmitted by vectors, especially in the context of climate changes.

## 5. Tendencies and Gaps

In the last decades, the *Aedes aegypti* vector has had more geographical reach. There is a tendency in studies to relate such geographical amplification with the mosquito's oviposition dynamics and associate it with its natural preference of establishing in different environments due to socio-environmental conditions, with emphasis on meteorological and basic sanitation variables (water supplying and methods of solid waste elimination, mainly). However, studies alert that random waste disposal is not an established risk factor for chikungunya, despite being related to an increase of the vector's breeding spots (intermediate host) and the aforementioned disposal results in scenarios that may favor the transmission of other pathogens such as bacteria, fungi and protozoa.

Besides that, studies have been incorporated in discussions about the association of *Aedes*, arbovirus diseases and environmental variables concerning the theme of community involvement in health promotion. This is an alert to humanity to unite our efforts in facing the challenges involving the prevention of vector-transmitted diseases, especially in the context of climate change.

However, the connection between the climate and the *Aedes aegypti* vector population must be investigated in large-scale studies before being used as preview models [54].

In regions where the disease is reappearing, such as dengue in Brazil, there is little knowledge about the possible differences in its epidemiology in epidemic and interepidemic scenarios, and this may hinder the identification of the determinant factors for its transmission [80].

Thus, regarding the predictor variables for the dissemination of arbovirus diseases (especially dengue, Zika and chikungunya), a better understanding can be reached in future studies with the inclusion of entomological investigation in the verification of relations between the environment and the circulation of pathogens and related vectors.

### 6. Conclusions

The present work allowed us to conclude that:

- The variable water permeates between the categories (SAN, CLIM and INT) and other research variables, showing notable relevance for public health, especially in the context of basic sanitation.
- The *Aedes aegypti* breeding spots and occurrence of diseases such as dengue, Zika and chikungunya are related to 16 variables, evaluated individually or jointly within one or more categories defined in this study.
- There are three categories that cover the located variables, these being basic sanitation, climatologic elements and secondary variables, which have been named "integrative" in this work.
- A total of 66% of the works have used some basic sanitation components in the calculation of correlations in this research topic, inferring that sanitation is a crucial indicator that can explain the proliferation of vectors in a given region and, consequently, the increase of incidence of arbovirus diseases.
- From 2010 to the time of this study, 16% of the works have incorporated discussions about *Aedes aegypti*, arbovirus diseases and environmental variables, the theme of community engagement in health promotion, alerting humanity to join efforts in facing challenges including the prevention of vector-transmitted diseases, especially in the context of climate changes.
- There is a demand for research involving a greater quantity of variables analyzed in tandem, which could better explain the dissemination or lack thereof of arbovirus diseases in a given location.
- In posterior studies, the group of 16 variables may contribute to the creation of indexes that would subsidize a better understanding of the health–disease process, in an analysis of the rural environment concerning measures to prevent diseases and promote health.

**Supplementary Materials:** The following supporting information can be downloaded at: https://www.mdpi.com/article/10.3390/su15108148/s1, title; Table S1: Socio-environmental variables found in the research and considered predictors for the incidence of Dengue, Zika, Chikungunya and/or for the proliferation of vectors of the genus *Aedes* through the abundance of breeding sites.

**Author Contributions:** Conceptualization, A.C.d.S. and P.S.S.; methodology, A.C.d.S. and P.S.S.; software, A.C.d.S. and P.S.S.; formal analysis, A.C.d.S. and P.S.S.; investigation, A.C.d.S.; resources, A.C.d.S.; data curation, A.C.d.S.; writing—original draft preparation, A.C.d.S.; writing—review and editing, A.C.d.S. and P.S.S.; visualization, A.C.d.S. and P.S.S.; supervision, P.S.S.; project administration, P.S.S.; funding acquisition, P.S.S. All authors have read and agreed to the published version of the manuscript.

**Funding:** This research was funded by Fundação Nacional de Saúde (FUNASA), TED 05/2017 and the APC was funded by Fundação Nacional de Saúde (FUNASA). This study was financed in part by the Coordenação de Aperfeiçoamento de Pessoal de Nível Superior—Brasil (CAPES)—Finance Code 001.

**Institutional Review Board Statement:** Not applicable.

**Informed Consent Statement:** Not applicable.

**Data Availability Statement:** Not applicable.

**Acknowledgments:** The Authors would like to thank everyone involved in the Projeto Saneamento e Saúde Ambiental em Comunidades Rurais e Tradicionais de Goiás—SanRural, for promoting scientific research in the state of Goiás. We would like to thank Maykell Guimarães for providing Figures 1 and 2.

**Conflicts of Interest:** The authors declare no conflict of interest.

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
