# Peer review of "Environmental Variables Related to Aedes aegypti Breeding Spots and the Occurrence of Arbovirus Diseases"

_sustainability, doi:10.3390/su15108148_

Round 1
Reviewer 1 Report
In this review essay, the author essentially assessed various environmental, social-democratic, and fundamental hygienic factors that contribute to the spread of various fevers carried on by the presence of mosquitoes, including dengue, yellow fever, etc. Based on the argument, “there has been an increase in the last few years in the number of people which find these arbovirus diseases in the rural environment putting at risk that population’s health. Thus the author is interested to know the predictor variables of breeding spots for the Aedes aegypti.
Overall, the paper summarizes the current state of knowledge of the breeding spots for the Aedes aegypti by keeping in view the different related variables.
1. In my view author should also mention the nature of the data set ( survey, time series, etc) used in reviewed articles along with sanitation (SAN), climatologic (CLIM), and integrative (INT).
2. I advise distributing the review to both developing and developed nations. We can get a clearer picture of related variables in this way. because the environmental and social-democratic characteristics vary from country to country.
3. 3. You can also categorize the research by geographic location, for example, Sri Lanka, India, and Nepal are all Asian countries and are more likely to share social-democratic values.

Author Response
In this review essay, the author essentially assessed various environmental, social-democratic, and fundamental hygienic factors that contribute to the spread of various fevers carried on by the presence of mosquitoes, including dengue, yellow fever, etc. Based on the argument, “there has been an increase in the last few years in the number of people which find these arbovirus diseases in the rural environment putting at risk that population’s health. Thus the author is interested to know the predictor variables of breeding spots for the Aedes aegypti.
Overall, the paper summarizes the current state of knowledge of the breeding spots for the Aedes aegypti by keeping in view the different related variables.
- In my view author should also mention the nature of the data set (survey, time series, etc) used in reviewed articles along with sanitation (SAN), climatologic (CLIM), and integrative (INT).
Answer: Supplementary material and new texts have been included throughout the article. We hope we have met the recommendation.
- I advise distributing the review to both developing and developed nations. We can get a clearer picture of related variables in this way. because the environmental and social-democratic characteristics vary from country to country.
Answer: It has been included in supplemental material and in Figure 5
- You can also categorize the research by geographic location, for example, Sri Lanka, India, and Nepal are all Asian countries and are more likely to share social-democratic values.
Answer: due to the amount of data obtained, this analysis was not possible, they are dispersed, but we include them in the texts of the article.

Reviewer 2 Report
Silva and Scalize conducted a systematic literature review of published works in bibliometrics databases, and conducted qualitative and quantitative analysis (statistical analysis) of result. The goal of this work was to identify and evaluate the pertinence of environmental variables that can allow the growth of Aedes aegypti breeding spots and the eventual increase of dengue fever, Zika and chikungunya in rural areas. It is helpful for concerning the prevention of diseases. But it still need promote. The comments are listed as following.
P1, L12: “Aedes aegypti” the format should be in italics, and "the" before it should be deleted
P1, L17: “conducted by” instead of “conducted”
P1, L18: “qualitatively” instead of “qualitative”
P1, L19: “of” instead of “from”
P1, L25: “on” instead of “in”
P1, L25: delete “,”
P1, L25: “on” instead of “in”
P1, L26: “these” instead of “theses”
P1, L27: “into” instead of “in”
P1, L27: “make” instead of “making”
P1, L27: “with” instead of “to”
P1, L27: “change” instead of “changes”
P2, L63: “vary” instead of “varies”
P2, L64-66: Word order error
P3, L81: “culicidae” instead of “culicidae’s”
P4, L87:“Owing to” or “In consequence of” instead of “Due to”
P4, L110: delete “in”
P4, L110: delete “which are”
P5, L147: “associations” instead of “association”
P5, L150: “Snyder’s” instead of “Snyder”
P5, L174: “of” instead of “from”
P5, L176: “vectors” instead of “vector”
P6, L192-193: The formatting of your Figure 4 is not standard, please remake it.For example, it lacks axes and error bars.In order to ensure the readability and accuracy of the data, axes and error bars are typically included in charts. If these elements are indeed missing, attempts can be made to correct the chart and add these elements to help readers better understand the data.
P8, L244-245: Figure 5 is not standard, please remake it.
P10, L268-269: Figure 7 is not standard, please remake it.
P11, L300: “during” instead of “where”
P11, L314: “indiscriminately” or “haphazardly” instead of “without criteria”
P11, L325: “with regard to” instead of “so to”
P12, L345: “who” instead of “which”
P13, L394: “rural areas” instead of “rural”
P13, L394: “pupae” instead of “pupas”
P13, L401: “have” instead of “has”
P13, L409: “enhance” instead of “enhances”
P13, L440: Add “by” after the “diseases”
P14, L459: “incorporated” instead of “incorporating”
Please pay attention to the same format of the first letter of the reference title.

Author Response
Review 2
Comments and Suggestions for Authors
Silva and Scalize conducted a systematic literature review of published works in bibliometrics databases, and conducted qualitative and quantitative analysis (statistical analysis) of result. The goal of this work was to identify and evaluate the pertinence of environmental variables that can allow the growth of Aedes aegypti breeding spots and the eventual increase of dengue fever, Zika and chikungunya in rural areas. It is helpful for concerning the prevention of diseases. But it still need promote. The comments are listed as following.
P1, L12: “Aedes aegypti” the format should be in italics, and "the" before it should be deleted- ok
P1, L17: “conducted by” instead of “conducted” - ok
P1, L18: “qualitatively” instead of “qualitative” - ok
P1, L19: “of” instead of “from” - ok
P1, L25: “on” instead of “in” - ok
P1, L25: delete “,”- ok
P1, L25: “on” instead of “in” - ok
P1, L26: “these” instead of “theses” - ok
P1, L27: “into” instead of “in” - ok
P1, L27: “make” instead of “making” - ok
P1, L27: “with” instead of “to” - ok
P1, L27: “change” instead of “changes” - ok
P2, L63: “vary” instead of “varies” - ok
P2, L64-66: Word order error: ok! => Therefore, in order to reduce the vector’s proliferation, the interruption of its life cycle via the elimination of breeding spots is necessary, and the aforementioned breeding spots [10,14,15] can be seen in places inhabited by humans [16,17].
P3, L81: “culicidae” instead of “culicidae’s” - ok
P4, L87: “Owing to” or “In consequence of” instead of “Due to” - ok
P4, L110: delete “in” - ok
P4, L110: delete “which are” - ok
P5, L147: “associations” instead of “association” - ok
P5, L150: “Snyder’s” instead of “Snyder” - ok
P5, L174: “of” instead of “from” - ok
P5, L176: “vectors” instead of “vector” - ok
P6, L192-193: The formatting of your Figure 4 is not standard, please remake it.For example, it lacks axes and error bars.In order to ensure the readability and accuracy of the data, axes and error bars are typically included in charts. If these elements are indeed missing, attempts can be made to correct the chart and add these elements to help readers better understand the data. Ok! figure redone and supplementary material included
P8, L244-245: Figure 5 is not standard, please remake it. 0k! figure redone and including continent and country
P10, L268-269: Figure 7 is not standard, please remake it. 0k! figure redone
P11, L300: “during” instead of “where” - ok
P11, L314: “indiscriminately” or “haphazardly” instead of “without criteria” - ok
P11, L325: “with regard to” instead of “so to” - ok
P12, L345: “who” instead of “which” - ok
P13, L394: “rural areas” instead of “rural” - ok
P13, L394: “pupae” instead of “pupas” - ok
P13, L401: “have” instead of “has” - ok
P13, L409: “enhance” instead of “enhances” - ok
P13, L440: Add “by” after the “diseases” - ok
P14, L459: “incorporated” instead of “incorporating” - ok
Please pay attention to the same format of the first letter of the reference title. - ok

Reviewer 3 Report
This manuscript is very interesting and intelligible. It focuses on the relationships of many variables of environment factors (sanitation, climatology and socio-environmental factors) and breeding spots of Aedes aegypti. This systemic review shows reasonable results and discussion.
These are some comments and suggestions:
1. Line 12, 46, 50 and 57, please edit “Aedes” as italic word.
2. Line 26, please check writing of “arbovirus”
3. Line 29, please check the word “predator”. p should not be bold.
4. Line 48 and 49, please check the spelling of scientific name.
5. Line 57, please clarify the white lines in figure 1.
6. Figure 2 No 1, please check format of scientific name.
7. Line 125, please clarify the criteria of the keywords that the author selected. How did the author select these keywords in search system?
8. Table 2, please separate result of %recurrence into 2 columns and give title of each column. What is the name of title category of 86 and 14%?
9. Line 307, please delete “(BARRERA et al., 1995)”.
10. Line 534, Reference 9 please delete “p”.
11. Line 586, Reference 29, DOI is not found.

Author Response
Comments and Suggestions for Authors
This manuscript is very interesting and intelligible. It focuses on the relationships of many variables of environment factors (sanitation, climatology and socio-environmental factors) and breeding spots of Aedes aegypti. This systemic review shows reasonable results and discussion.
These are some comments and suggestions:
- Line 12, 46, 50 and 57, please edit “Aedes” as italic word. - ok
- Line 26, please check writing of “arbovirus” - ok
- Line 29, please check the word “predator”. p should not be bold. - ok
- Line 48 and 49, please check the spelling of scientific name. - ok
- Line 57, please clarify the white lines in figure 1. – ok => Note: The White lines illustrate the possible random routes of the Aedes vector from one place to another.
- Figure 2 No 1, please check format of scientific name. - ok
- Line 125, please clarify the criteria of the keywords that the author selected. How did the author select these keywords in search system? Ok! The choice of keywords is based on the goal of reaching the biggest possible amount of works which involved components of basic sanitation (water, waste, swage and drainage) and climatology (generally, meteorological elements) in association with arbovirus and Aedes.
- Table 2, please separate result of %recurrence into 2 columns and give title of each column. What is the name of title category of 86 and 14%?
Answer: It was referring to the sum of the percentages in the left column, but we decided to remove it because it didn't add anything.
- Line 307, please delete “(BARRERA et al., 1995)” - ok
- Line 534, Reference 9 please delete “p”. - ok
- Line 586, Reference 29, DOI is not found. - ok
Materials and methods
- What criteria did the author select the keywords in search system using the Stings?
Answer: The choice of keywords is based on the goal of reaching the biggest possible amount of works which involved components of basic sanitation (water, waste, swage and drainage) and climatology (generally, meteorological elements) in association with arbovirus and Aedes.
- There are some confusing about the pertinence to the theme, please describe more details about "non-relevant (N), relevant (S), review pertinence (R)".
Answer: That way, in Step 2, from the surveyed productions selected in Step 1, the works were analyzed regarding their titles in order to exclude those inadequate to the theme (named “N”, regarding socio-environmental variables’ not pertinent to arbovirus and vectors) and the duplicate/overlapping ones. Where the ones classified as “S” were immediately reserved for integral reading. Sequentially, on step 3, the works from Step 2 classified as “R” were reviewed concerning their pertinence and analyzed through the reading of their abstracts to classify them as pertinent or not, as the mere title information did not clarify the research question. The ones tagged as “S” were sent forward to Step 4, that is, those whose theme referred to the studies regarding association involving environmental variables, Aedes aegypti breeding spots and the prevalence or incidence of arbovirus diseases.
Discussion Please give more details of application according to the author's result. How do these variables (SAN, CLIM and INT) relating to Aedes aegypti breeding spots involve with arbovirus disease incidence and transmission?
Answer: we have included supplementary material and new texts have been included throughout the article. We hope to have met the demand
Gaps
The authors informed and discussed more about the environmental factors relating to Aedes mosquitoes. However there are other factors eg, hosts and pathogens that relate to the environment. Could you please explain more details or gaps of hosts, pathogens and vectors that relate to environment in term of disease transmission and control
Answer:
In the last decades, the Aedes aegypti vector has had more geographical reach. There is a tendency in studies to relate such geographical amplification with the mosquito’s oviposition dynamics, and associating to its natural preference of establishing in different environment due to socio-environmental conditions, with emphasis to meteorological and basic sanitation variables (water supplying and methods of solid waste elimination, mainly). However, studies alert that the random waste disposal is not an established risk factor for chikungunya, despite being related to the increase of the vector’s breeding spots (intermediate host), and the aforementioned disposal results in scenarios which may favor the transmission of other pathogens such as bacteria, fungi and protozoa.
Besides that, studies have been incorporated in discussions about the association of Aedes, arbovirus diseases and environmental variables the theme regarding community involvement in health promotion. That is an alert to humanity to unite efforts in facing the challenges involving the prevention of vector-transmitted diseases, especially in the context of climate changes.
However, the connection between climate and Aedes aegypti vector population must be investigated in large-scale studies, before being used as preview models [54].
In regions where the disease is reappearing, such as dengue in Brazil, there is little knowledge about the possible differences in its epidemiology in epidemic and inter-epidemic scenarios, and this may hinder the identification of the determinant factors for its transmission [80].
Thus, regarding the predictor variables for the dissemination of arbovirus diseases (especially dengue, Zika and chikungunya), a better understanding can be reached in future studies with the inclusion of entomological investigation in the verification of relations between the environment and the circulation of pathogens and related vectors.
Reviewer 4 Report
1. Introdution
Line 48. Put the name of the specific epithet of Aedes albopictus and Aedes aegypti with the initial letter in lower case.
2. Materials and Methods
A suggestion would be to explain which characteristics caused the papers to be classified as non-relevant (N) in step one. What criteria do you use?
3. Results
In table 2 it is necessary to align the results in the indicator gG/IgM/ELISA, NS1 and/or RT-PCR serology the Recurrence value is misaligned. And about that it is not possible to understand why there are 2 columns in Recurrence (value 86).
4. Discussion
Line 359. Correct the word work that started with a capital letter.
Line 366. Suggestion would be to put a full stop after disease and start another sentence.
Author Response
Comments and Suggestions for Authors
- Introdution
Line 48. Put the name of the specific epithet of Aedes albopictus and Aedes aegypti with the initial letter in lower case.
Answer: done
- Materials and Methods
A suggestion would be to explain which characteristics caused the papers to be classified as non-relevant (N) in step one. What criteria do you use?
Answer: That way, in Step 2, from the surveyed productions selected in Step 1, the works were analyzed regarding their titles in order to exclude those inadequate to the theme (named “N”, regarding socio-environmental variables’ not pertinent to arbovirus and vectors) and the duplicate/overlapping ones. Where the ones classified as “S” were immediately reserved for integral reading. Sequentially, on step 3, the works from Step 2 classified as “R” were reviewed concerning their pertinence and analyzed through the reading of their abstracts to classify them as pertinent or not, as the mere title information did not clarify the research question. The ones tagged as “S” were sent forward to Step 4, that is, those whose theme referred to the studies regarding association involving environmental variables, Aedes aegypti breeding spots and the prevalence or incidence of arbovirus diseases.
- Results
In table 2 it is necessary to align the results in the indicator gG/IgM/ELISA, NS1 and/or RT-PCR serology the Recurrence value is misaligned. And about that it is not possible to understand why there are 2 columns in Recurrence (value 86).
Answer: It was referring to the sum of the percentages in the left column, but we decided to remove it because it didn't add anything.
- Discussion
Line 359. Correct the word work that started with a capital letter. - ok
Line 366. Suggestion would be to put a full stop after disease and start another sentence. - ok

Round 2
Reviewer 1 Report
draft has been improved. author carefully incorporates all comments.
Author Response
Thanks